# Estimating subnational excess mortality in times of pandemic. An application to French *départements* in 2020

**Florian Bonnet**[ID][☋]*, **Carlo-Giovanni Camarda**[ID][☋]

Institut national d'études démographiques (INED), Aubervilliers, France

☋ These authors contributed equally to this work.
* florian.bonnet@ined.fr

**Data Availability Statement:** See the open source framework repository https://osf.io/zt2c8/.

**Funding:** The authors received no specific funding for this work.

## Abstract

The COVID-19 pandemic's uneven impact on subnational regions highlights the importance of understanding its local-level mortality impact. Vital statistics are available for an increasing number of countries for 2020, 2021, and 2022, facilitating the computation of subnational excess mortality and a more comprehensive assessment of its burden. However, this calculation faces two important methodological challenges: it requires appropriate mortality projection models; and small populations imply considerable, though commonly neglected, uncertainty in the estimates. We address both issues using a method to forecast mortality at the subnational level, which incorporates uncertainty in the computation of mortality measures. We illustrate our approach by examining French départements (NUTS 3 regions, or 95 geographical units), and produce sex-specific estimates for 2020. This approach is highly flexible, allowing one to estimate excess mortality during COVID-19 in most demographic scenarios and for past pandemics.

## 1 Introduction

Estimating COVID-19 mortality has been the object of intense research, both to guide public policies aimed at curbing the spread of the virus and to determine the pandemic's global burden in various countries. National health surveillance agencies were first mobilized to provide weekly or even daily COVID-19 death tolls and thus establish a rapid indicator of the pandemic's impact [1, 2]. However, differences in data definitions among countries, time-varying collection methods, reporting delays, and inconsistent coverage by place of death are known issues that impede the use of health surveillance systems for reliably assessing the pandemic [3, 4].

Over the months, official statistics systems have provided information that complements and/or corrects surveillance-system data, including deaths by age from all causes. These data are the basis for constructing excess mortality measures to more comprehensively assess the pandemic's burden. Defined as "the difference between the number of deaths (from any cause) that occur during the pandemic and the number of deaths that would have occurred in the absence of the pandemic" [5, p. 85], excess mortality can also be applied to other indicators like life expectancy and standardized death rates. Measures of excess mortality have been

**Competing interests:** The authors have declared that no competing interests exist.

considered the gold standard for estimating the impact of COVID-19 [6, 7], and they have been exten-sively adopted in the past years. Whereas some of these measures have used pre-pandemic years as baseline mortality in the absence of COVID-19 [8, 9], others have accounted for mortality changes over time [10–14].

However, all these studies have estimated excess mortality at the national level. This perspective generally obscures large regional differences that ought to be taken into account to better inform policymakers. Hence, many recent studies have attempted to estimate excess mortality at the regional level. For some of these, excess mortality is the difference between regional mortality levels in 2020 or 2021 and pre-pandemic mortality. The countries thus analyzed have been Brazil [15], Italy [16, 17], Mexico [18], Portugal [19], Spain [20], Sweden [21], Switzerland [22] and the United States [23]. More thoughtful accounting of the temporal change in mortality via forecasting techniques have been also proposed for estimating excess mortality at regional level. While various countries have been analyzed [24]studies focusing on specific countries have used different methods and pursued distinct objectives. Examples encompass Belgium, [25], Ecuador [26], England and Wales [27], Italy [28, 29], Latvia [30], Thailand [31], and United States [32].

While the value of producing excess mortality measures at a fine geographic scale seems clear and timely, the methodological challenges are numerous and often neglected. They are essentially related to small populations that naturally display high stochastic variation in death counts. Possible interpretations of regional differences are necessarily limited, but what becomes crucial are robust, flexible, and efficient methods for forecasting mortality levels in the theoretical absence of a pandemic, as well as for computing uncertainty in estimates. Concerning the first issue, estimating baseline mortality in the absence of COVID-19 by extrapolating pre-pandemic trends is crucial for two primary reasons. In general, assuming a static pre-pandemic level as the counterfactual scenario can be overly restrictive. Moreover, at the subnational level, pre-pandemic mortality may exhibit considerable volatility, making the subjective decision of selecting the number of pre-pandemic years as the baseline particularly challenging.

Our primary aim is thus to tackle all these challenges within a clear-cut and unified framework. To achieve this, we present a novel approach for estimating subnational excess mortality during pandemics. In essence, our approach is designed to handle all the mentioned issues and provide estimates and uncertainty quantification of the pandemic's burden, regardless of the chosen metric for measuring excess mortality and the geographical granularity of the available data.

Specifically, we use *CP*-splines [33] to project mortality, since this approach exhibits two relevant features when dealing with small area mortality analysis: high flexibility in modeling diverse demographic scenarios comes along with robustness with respect variation due to small populations at risk. To obtain reliable measures of uncertainty around estimates for the numerous subpopulations at hands, we present an efficient analytic procedure that offers significant advantages in terms of computational cost and time.

To illustrate this approach, we compute sex-specific excess mortality in 2020 for the 95 départements (departments) in metropolitan France, which correspond to the third level of the Nomenclature of Territorial Units for Statistics (NUTS 3) used by Eurostat. In order to make comparisons and emphasize the significance of the various sources of uncertainty, we have conducted additional analysis at the NUTS 2 level (régions). This allows us to further explore and highlight the differences in uncertainty across different geographical granularity. While providing specific outcomes for France, the larger aim of this paper is to provide a general framework for computing excess mortality and associated uncertainty at the subnational level. Given this purpose, R routines [34] are publicly available to replicate our methodology

for other countries and historical contexts. See the open source framework repository on this link https://osf.io/zt2c8/.

## 2 Methods

To calculate excess mortality, we consider historical mortality trends and age-patterns in each subnational population, and do so separately for males, females, and both sexes combined. We also compute associated uncertainty using a simple, albeit rigorous, procedure, allowing us to separate the main sources of variation before assessing any possible mortality shock in small areas. For simplicity, we focus on 2020. However, the baseline mortality in the absence of the pandemic for successive years can be estimated by extending the forecast horizon of the model illustrated below. As a result, if subnational level data are accessible for 2021 and 2022, measuring excess mortality and quantify its uncertainty for these years can be achieved by simply adapting our approach.

For a given subpopulation and sex, we have $\mathbf{D} = (d_{ij})$ and $\mathbf{N} = (n_{ij})$, $m \times n$ matrices of deaths and exposures. We define exposures ($N$) as the mean of populations at 1st January for two consecutive years which is a suitable approximation of population exposed to the risk of death during a single age-time interval. Number of deaths $d_{ij}$ at age $i$ in year $j$ are assumed to be realizations from a Poisson distribution with mean $\mu_{ij}n_{ij}$ [35], where $\mu_{ij}$ is commonly named force of mortality. To compute a theoretical level where the pandemic had not occurred, we model observed mortality for pre-pandemic years (up to 2019) and forecast $\mu_{ij}$ for 2020. We adopt a *CP*-spline model [33] for forecasting mortality in 2020. This method enables the simultaneous estimation and forecast of mortality within a regression setting. Its main advantage lies in a single variance-covariance matrix that encompass uncertainty on both past and future mortality. Let arrange the complete matrices as a column vector, that is, $\mathbf{d} = \text{vec}(\mathbf{D})$ and $\mathbf{n} = \text{vec}(\mathbf{N})$. Mortality over all ages and years can thus be expressed as the exponential of a linear combination of *B*-spline basis and coefficients:

$$\boldsymbol{\mu} = \exp(\mathbf{B}\,\boldsymbol{\alpha})\,, \tag{1}$$

where $\mathbf{B}$ a two-dimensional model matrix that combines *B*-splines over age and years. In simple terms, $\mathbf{B}$ is the result of a (Kronecker) product of two sets of equidistant *B*-splines, built separately over age and time. This construction gives rise to a dense pattern resembling an "egg carton," where each hill has an associated coefficient. The estimation process follows a classic regression Poisson setting, but with an additional discrete penalty to ensure smoothness of the coefficient vector $\boldsymbol{\alpha}$ and, consequently, smoothness of mortality $\boldsymbol{\mu}$. In addition to this framework, *CP*-splines impose constraints on $\boldsymbol{\alpha}$ for future years to align future 2020 mortality within shapes estimated from pre-pandemic years. A comprehensive description of *CP*-splines is provided in [33].

A crucial factor is the selection of the most appropriate period for applying the mortality forecasting model. Rather than using all the accessible data or a uniform common year for all regions, we optimize the time-window for each region. We apply *CP*-splines with a rolling starting year up to 2010, then forecast 2019 and measure the distance between the observed and forecasted 2019 mortality. Working in a Poisson setting, we opt to measure distance by deviance [36, p. 34]. The starting year with the lowest deviance value was selected for the final analysis.

A measure of excess mortality for 2020 is defined as the difference between the value of a demographic indicator in a theoretical baseline mortality level and the value for the same indicator obtained from the observed mortality. Whereas the former is obtained by *CP*-splines and

solely dependent on the estimated coefficients $\boldsymbol{\alpha}$ in (1), observed death rates $\boldsymbol{m}_{2020} = \boldsymbol{d}_{2020}/\boldsymbol{n}_{2020}$ are the bases for computing actual level of mortality in the pandemic year.

For simplicity, we illustrate the procedure for calculating excess mortality measured by life expectancy at birth, $e_0$, but the entire process can be customized and applied to other demographic indicators. The point estimate of excess mortality is obtained by subtracting the observed life expectancy at birth from the forecasted value. In the following matrix formulation:

$$\delta_{e_0} = e_0^F(\boldsymbol{\alpha}) - e_0^O(\boldsymbol{m}_{2020}) = \mathbf{1}'_m[\exp(\mathbf{C}\,\mathbf{L}\,\boldsymbol{\mu}) - \exp(\mathbf{C}\,\boldsymbol{m}_{2020})]\,, \tag{2}$$

where $\mathbf{C}$ is a $m \times m$ lower-triangular matrix with -1 entries for computing the cumulative summation of mortality, e.g. $\boldsymbol{Cm}_{2020}$ corresponds to the discrete counterpart of minus the cumulative observed hazard function in 2020. The matrix $\boldsymbol{L}$ is constructed in order to select 2020 forecast mortality from (1). Its explicit form is presented in the S1 Appendix. The $m \times 1$ matrix of 1s, $\mathbf{1}_m$, serves to sum up the exponential of minus the cumulative hazards over all ages. A similar matrix approach for computing life expectancy has been proposed in the literature [37]. Furthermore, it is important to note that Eq (2) can be applied regardless of the forecasting approach used to obtain rates $\boldsymbol{\mu}$ (see comparative analysis in S2 Appendix).

Besides their estimated values, both observed and forecasted values for any mortality measure embody levels of uncertainty to be accounted for before drawing any conclusions about their change. This consideration is particularly relevant in subnational analyses when examining relatively small populations at risk. Instead of following time-consuming simulation and bootstrap procedures, we develop an analytic construction of the variance associated with both observed and forecasted mortality indicators by using delta method.

Unlike other methods, using *CP*-splines offers the advantage of providing an analytical expression for the variance–covariance matrix, cov$\boldsymbol{\alpha}$ [38 p. 32–33]. This matrix is dependent on the estimated coefficients $\boldsymbol{\alpha}$ and can be used to quantify the uncertainty associated with the forecasted life expectancy at birth ($e_0^F$). However, the overall uncertainty related to Eq (2) also relies on the observed mortality level and can be represented by a diagonal matrix consisting of the inverse of observed deaths, cov$(\boldsymbol{m}_{2020}) = \texttt{diag}(1/\boldsymbol{d}_{2020})$.

The variance of $\delta_{e_0}$ can then be calculated using the delta method with ease. We make the assumption that $e_0^F(\boldsymbol{\alpha})$ and $e_0^O(\boldsymbol{m}_{2020})$ are independent variables that follow an asymptotic normal distribution. The variance in (2) can thus be determined by adding the variances associated with both the forecasted and observed life expectancy at birth in 2020:

$$\begin{aligned} \mathbb{V}[e_0^F(\boldsymbol{\alpha})] &= \nabla e_0^F(\boldsymbol{\alpha})\, \text{cov}(\boldsymbol{\alpha})\, \nabla e_0^F(\boldsymbol{\alpha})' \\ \mathbb{V}[e_0^O(\boldsymbol{m}_{2020})] &= \nabla e_0^O(\boldsymbol{m}_{2020})\, \text{cov}(\boldsymbol{m}_{2020})\, \nabla e_0^O(\boldsymbol{m}_{2020})'\,, \end{aligned} \tag{3}$$

where symbol $\nabla$ denotes the vector differential operator, i.e. the partial derivatives of life expectancy with respect to either $\boldsymbol{\alpha}$ or $\boldsymbol{m}_{2020}$. Expressing the partial derivatives of the forecasted and observed $e_0$ without making excessive simplifications, we have the following formulations:

$$\begin{aligned} \nabla e_0^F(\boldsymbol{\alpha}) &= \mathbf{1}'_m\, \texttt{diag}(\exp(\boldsymbol{CL}\mu))\, \boldsymbol{CL}\, \texttt{diag}(\mu)\, \boldsymbol{B} \\ \nabla e_0^O(\boldsymbol{m}_{2020}) &= \mathbf{1}'_m\, \texttt{diag}(\exp(\boldsymbol{Cm}_{2020}))\, \boldsymbol{C}\, \texttt{diag}(\boldsymbol{m}_{2020})\,. \end{aligned} \tag{4}$$

By employing this approach, we can effectively identify and separate the extent of uncertainty attributed to the estimated baseline mortality (hereafter: "*forecast uncertainty*") and the uncertainty arising from the observed mortality level in 2020 (hereafter: "*Poisson uncertainty*").

In S1 Appendix, we provide derivations to obtain excess mortality estimates and their associated uncertainty when measured by life expectancy at any given age x, age-standardized

death rates, and death toll. Additionally, S2 Appendix includes a comparison between the results obtained from the *CP*-splines and the more conventional Lee-Carter model [39]. This additional analysis provides reassurance regarding the robustness of the majority of the outcomes. When significant discrepancies between the two models were observed in estimating excess mortality, the *CP*-splines demonstrated superior performance in both modeling and forecasting mortality in small geographical units.

## 3 Results

To illustrate, we present excess mortality measures by losses/gains in life expectancy at age 60 ($e_{60}$) in 2020 for 95 *départements* of metropolitan France. The French Human Mortality Database [40] provides annual deaths (**D**) and population on January 1 by single age at death (with an open age interval 95+), sex, and *département* for each year between 1970 and 2020. In our figures, we specifically highlight male excess mortality, as it was observed that males were more significantly affected by the pandemic compared to females [41]. However, we have included outcomes for females in the S3 Appendix. This allows for a more comprehensive understanding of the gender-specific impact of the pandemic on mortality.

Fig 1 illustrates our methods and presents losses in male $e_{60}$ for a single subpopulation, *Territoire de Belfort*. This *département* was not chosen randomly. Strongly affected by the pandemic, it is a relatively small area (about 70,000 men in 2020) and therefore may show more variability in mortality due to the smaller sample size.

The upper panels of Fig 1 reveal how the forecasts include uncertainty around estimates. Whereas observed life expectancy at age 60 in 2020 was 21.41 years, our projected value lies

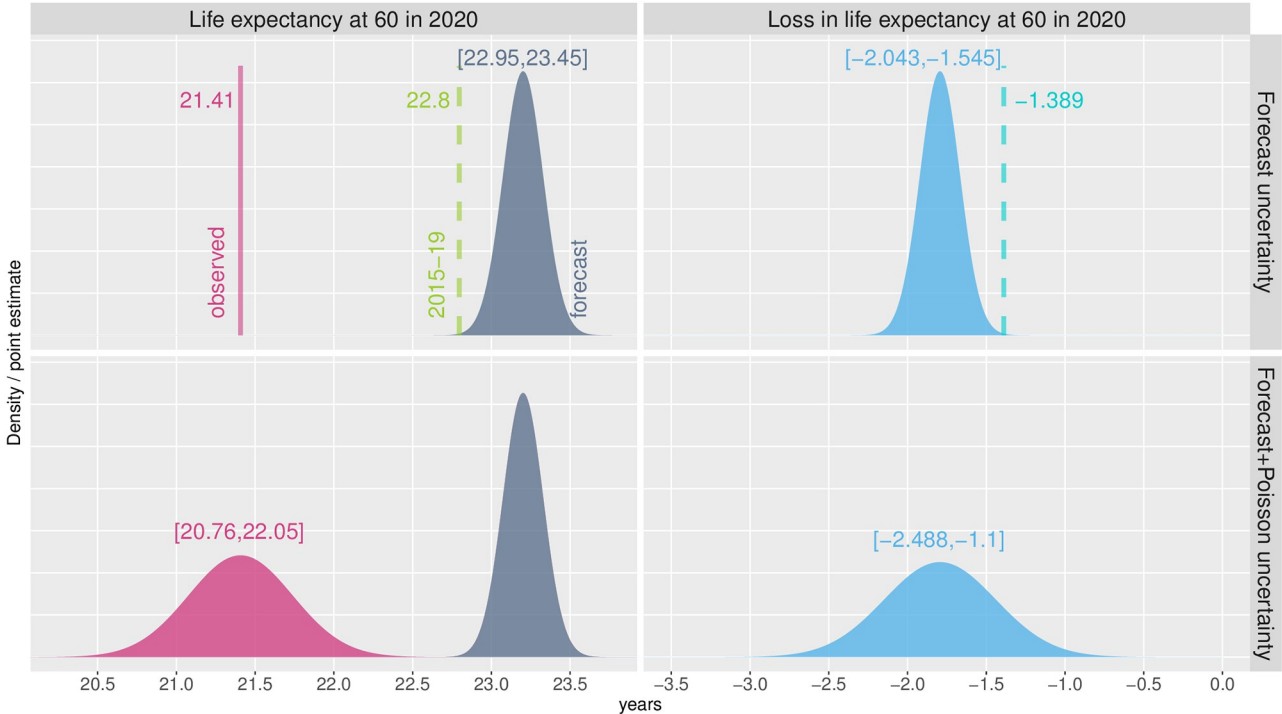

**Fig 1. Illustrative figure of sources of uncertainty around excess mortality measure.** Life expectancy at 60 (left panels) and associated losses (right panels) for *Territoire de Belfort*, males, 2020. Upper panels: forecast uncertainty is accounted. Lower panel: both forecast and Poisson uncertainty are reported. Texts refer to either point estimates or 95% confidence intervals. Dashed lines depict "simple" estimation of excess mortality.

between 22.95 and 23.45 years. Consequently, a loss in $e_{60}$ is estimated between approximately 1.5 and 2 years. For comparison, Fig 1 presents what we labelled as the "simple" estimate of excess mortality, used in many recent studies: the mortality level from the 5 pre-pandemic years is used as the theoretical baseline level without the pandemic. Ignoring decreasing trends in mortality, this approach biases the excess mortality estimate downward: here, the loss in $e_{60}$ is only 1.4 years.

The lower part of Fig 1 presents the Poisson uncertainty associated with the observed mortality level in 2020. Negligible when the population is large, this source of uncertainty becomes relevant at the regional level. In our example, adding the Poisson uncertainty around our estimates increases the confidence interval by 0.9 years. Thus, in 2020 and for *Territoire de Belfort*, we measure a loss in male $e_{60}$ between 1.1 and 2.5 years.

An often overlooked phenomenon deserves attention here. The uncertainty associated with observed mortality is greater than the variability around the forecasted mortality in 2020 (see bottom-left panel in Fig 1). While the former relies on the observed mortality for a particular year, the forecasted mortality is determined by a model that factors in information over both age and time, and prediction is done for a single year only. As the forecasting horizon extends, the uncertainty associated with it will possibly surpass the uncertainty associated with the eventual observed mortality. This phenomenon is independent of the specific forecasting method used.

In S3 Appendix, we replicate Fig 1 for *Seine-Saint-Denis*. This *département* was also heavily affected by the pandemic in 2020, but the male population is 13 times larger. Total uncertainty is thus much lower (0.8 years), with an estimated loss in $e_{60}$ between 2.4 and 3.2 years.

Interestingly, due to reduced gains in life expectancy since the beginning of the 2000s, the time-window used to forecast male mortality in 2020 starts between 2005 and 2010 for 56 out of 95 *départements*, and between 2000 and 2004 for 12 others.

Fig 2 presents point estimates of losses/gains in male $e_{60}$ for each *département* with their 95% confidence interval as well as "simple" estimates. Both sources of uncertainty, forecast and Poisson, are displayed. Fig 3 mirrors this information in a map of France: point estimates are displayed and *départements* with non-significant result at the 5% level are highlighted. Fig 2 and 3 of S3 Appendix replicate these figures for females.

For mainland France, we estimate that $e_{60}$ has decreased by 0.77 years, whereas "simple" estimate is almost twice lower (0.42). The uncertainty around this value is 0.15 years and mostly due to forecast; the Poisson uncertainty practically disappears when we deal with the whole French male population (33 million). Loss in male $e_{60}$ remarkably varies across subpopulations: the maximum loss was in *Seine-Saint-Denis* (2.4 years) whereas the minimum was in *Gers* (gain of 0.6 years). However, 95% confidence intervals around this point estimate are wide (1.2 years), resulting in a non significant gain at the 5% level.

Overall 26 *départements* show estimates that are not significant at the 5% level; only when losses in $e_{60}$ rise to about 0.4 years do we start to detect significant excess mortality, except for highly populated areas such as *Hérault* and *Loire-Atlantique*.

Fig 3 reveals the geography of the pandemic in 2020. Whereas Western France largely spared by the pandemic, estimates were larger in the east and in the *Ile-de-France* (Greater Paris region), with losses in life expectancy at age 60 about 1 and 1.5 years, respectively. These results are consistent with those from the French national statistical office (INSEE) [42, 43], which compared deaths observed in 2020 during the first two waves of the pandemic (March–April 2020 and September–December 2020) with those observed during the same periods in 2019. The first wave of the pandemic had strongly affected Greater Paris region, as well as the north and northeast of the country, probably due to the onset of the first outbreaks in mainland France in these regions. One of the first outbreaks was detected in the *Oise département*,

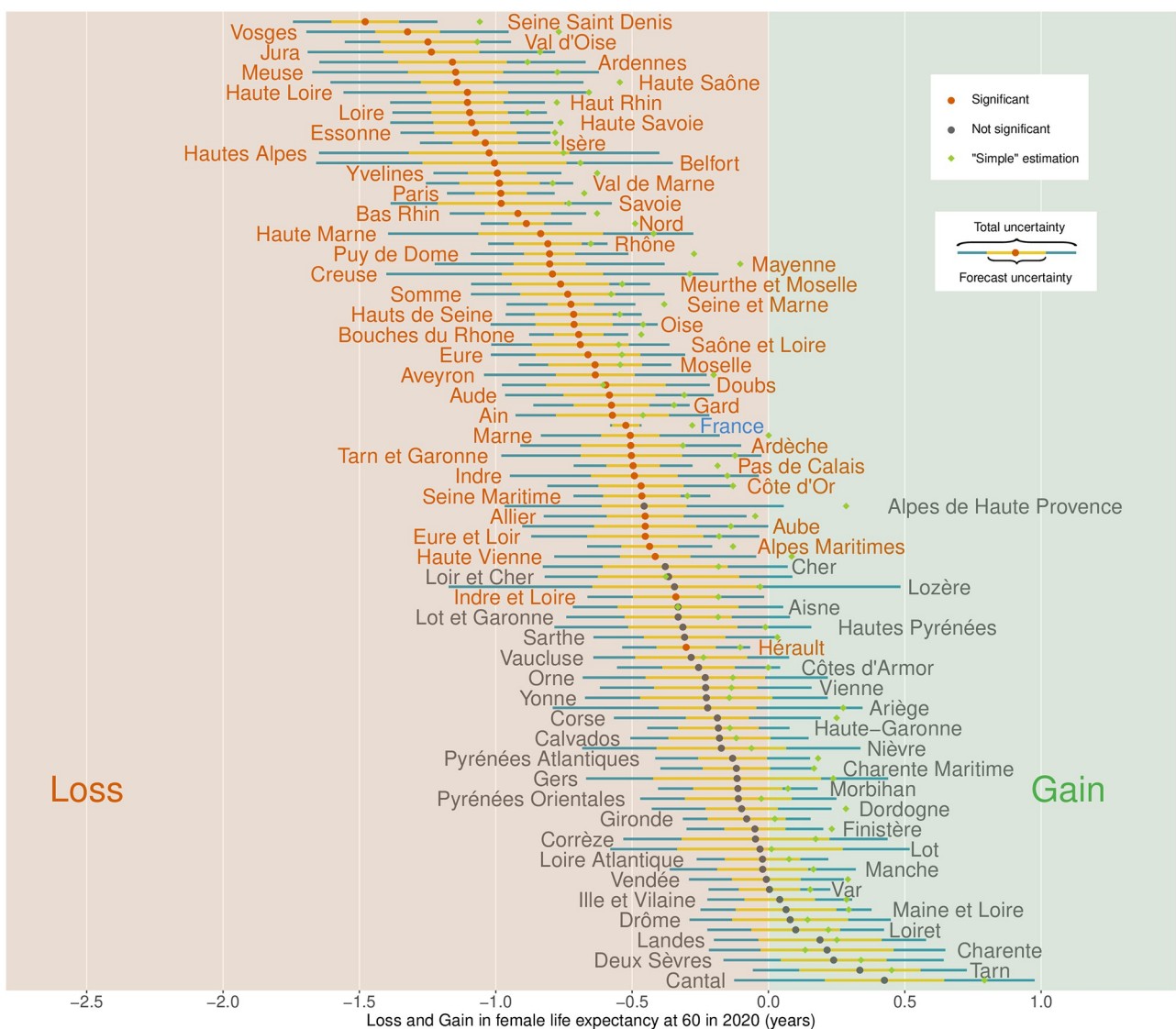

**Fig 2. Losses in male life expectancy at age 60 in 2020 for each French *département*.** Colors of dots and texts depict the presence of significant estimates at 95% level, and colors of the horizontal bars represent the two sources of uncertainty. Green dots identify "simple" estimates of losses.

in the immediate surroundings of the Charles de Gaulle airport (23 km northeast of Paris). Another outbreak was detected in *Haut-Rhin* and spread to the rest of northeastern France. The INSEE studies also showed that the second wave of the pandemic was strong throughout eastern France and particularly severe in the *Auvergne-Rhône-Alpes* region and neighboring *départements*.

A comparison of these results with those for females highlights that the loss in $e_{60}$ for females is lower than for males. At national level, we estimate that the loss in $e_{60}$ was 6 months, compared with 9 months for males. In detail, no *département* suffered from a loss in $e_{60}$ greater than 1.5 years for females, while 8 *départements* suffered such a loss for males.

For completeness, Tabs 1–3 in S3 Appendix include values of excess mortality estimates measured by $e_{60}$ for males, females and both sexes combined.

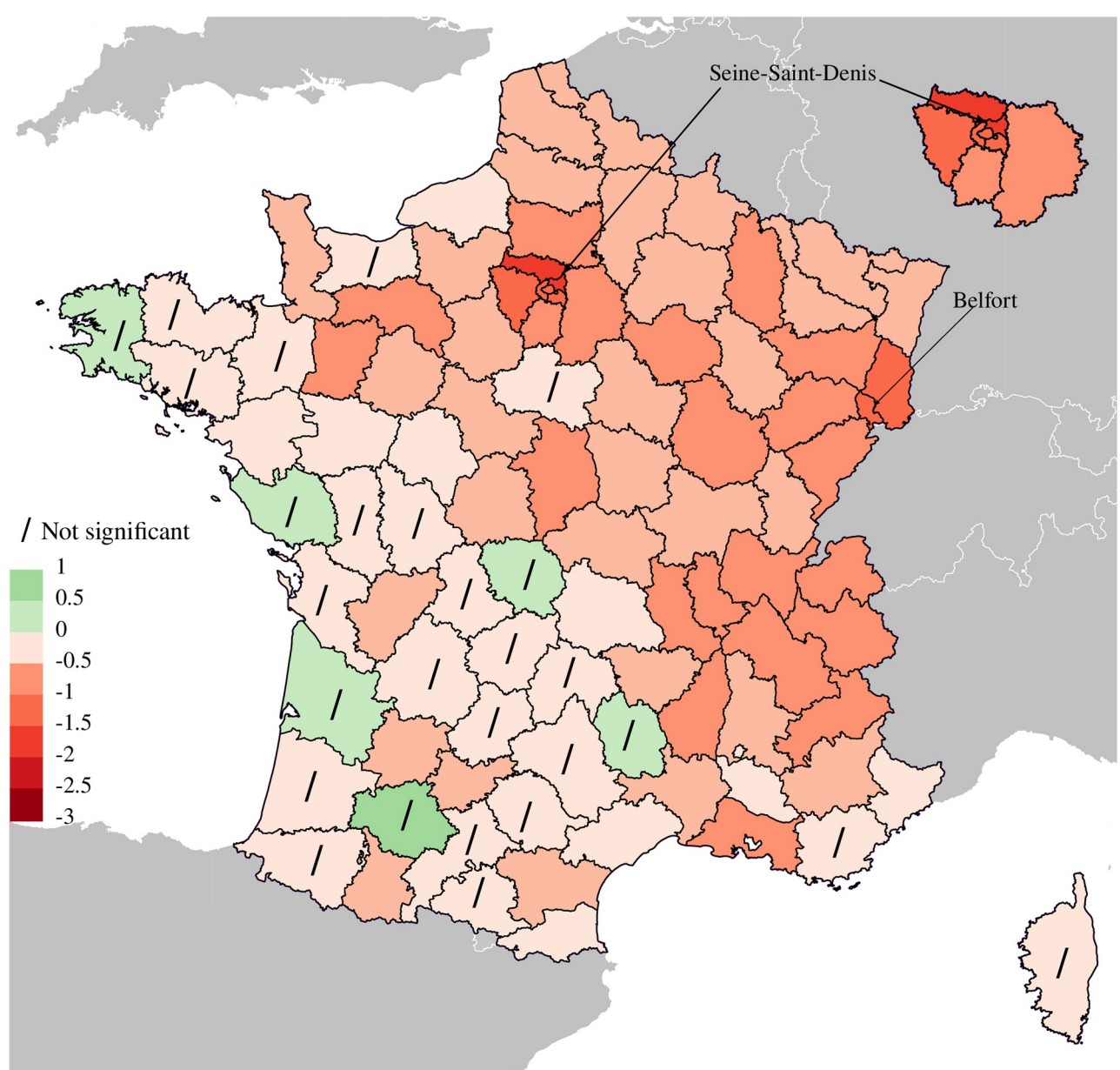

**Fig 3. Map of France *département* by losses/gains in male life expectancy at age 60 in 2020.** Slash symbol (/) denote areas with loss/gain in $e_{60}$ not significant at 5% level. On the upper-right corner zoom of a part of the map referring to Greater Paris (*Ile-de-France*).

A central aspect of this paper concerns the importance of measuring uncertainty around excess mortality estimates when dealing with small areas. Fig 4 is a log–log plot of the amount of uncertainty (measured by the range of the 95% confidence intervals) against population size, i.e. we illustrate the proportional change in uncertainty in response to a proportional change in population size. To broaden the view, we show both *départements* (NUTS 3) and *régions* (NUTS 2) and, along with the total uncertainty (in green), we differentiate uncertainty arising from the forecast procedure (in purple) and from Poisson on observed data (in orange).

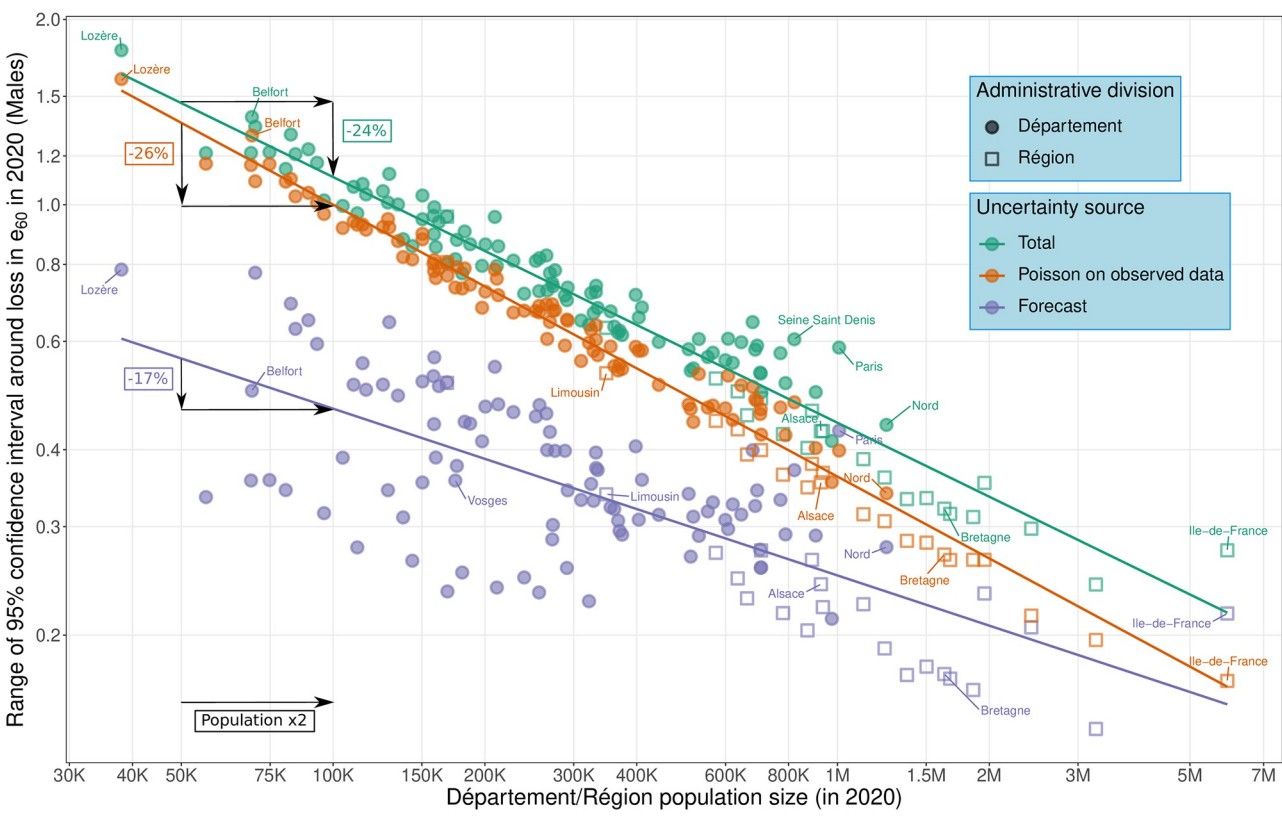

**Fig 4. Log-log plot of the amount of uncertainty against population size by source of uncertainty: Total, associated to the forecast process and due to Poisson randomness in observed data.** Uncertainty is measured by the width of the 95% confidence intervals around estimated loss in male $e_{60}$. Values for both *départements* (NUTS 3) and *régions* (NUTS 2) are depicted. Linear fits are provided for illustrative purposes and for obtaining an approximated value of the elasticity associated to each source of uncertainty.

With a linear fit to both departmental and regional values, we can estimate elasticity associated with source-specific uncertainty and gauge the percent change in uncertainty for a doubling in population size. We estimate that uncertainty decreases by 24% when the population doubles. This value combines two sources of uncertainty that decrease at an unequal pace when population grows. Whereas Poisson uncertainty is higher than the forecast uncertainty for almost all subnational populations, this source of uncertainty decreases at faster pace (26%) than the forecast uncertainty (17%) when the population doubles.

We can also read Fig 4 from an alternative perspective. When excess mortality is measured by $e_{60}$, a loss greater than 0.75 years would be necessary to have a significant estimate at the 95% level if the population size is 50,000. This value decrease with larger populations: in a region with 200,000 (one million) men, a loss in $e_{60}$ equal to 0.4 (0.2) would be required to obtain a significant excess mortality.

## 4 Discussion

Assessing excess mortality during a pandemic, such as COVID-19, is as crucial from a health policy perspective as it challenging from a methodological standpoint. The challenges increase when estimating excess mortality at subnational level. Specifically, we face two main issues.

First, mortality levels that would have been observed had pandemic not occurred need to be estimated. Thus, forecasting methods are necessary to extrapolate temporal mortality

variations. In this paper, we use *CP*-splines [33] and illustrate our approach with a reproducible example by examining French NUTS 3 regions. If data on deaths and exposure populations are available by age and year, this flexible approach is adaptable to a large variety of current and historical scenarios, and is robust for dealing with very small populations.

Second, when examining sparsely populated areas, the level of uncertainty increases significantly. In this paper, we calculate the uncertainty surrounding point estimates and differentiate between the uncertainty arising from the forecasting process and the inherent uncertainty stemming from the Poisson random nature of observed mortality data. We show that overall uncertainty in excess mortality decreases by 24% when the population doubles, though Poisson uncertainty tends to decrease more rapidly when the population grows. Consequently, while it is possible to safely ignore uncertainty in the observed data and focus solely on forecasting errors for large populations, it is crucial to consider the Poisson component of uncertainty when analyzing excess mortality in small areas. Before drawing any conclusions, one must account for the Poisson component to accurately assess the uncertainty associated with excess mortality.

One way to reduce uncertainty, namely the Poisson component, is to either gather populations spatially by aggregating smaller administrative divisions into larger ones, or estimate excess mortality for both sexes. Still, these choices must be made with caution because associated outcomes might hide strong heterogeneity.

To illustrate these concepts [Fig 5] shows the densities of losses in male $e_{60}$ for two specific French *régions* (lower panel) and associated lower-level administrative divisions, *départements* (upper panel). In this example, *Ile-de-France* or Greater Paris is the *région* that suffered from the highest loss in life expectancy at age 60, with a 95% confidence interval loss in $e_{60}$ of

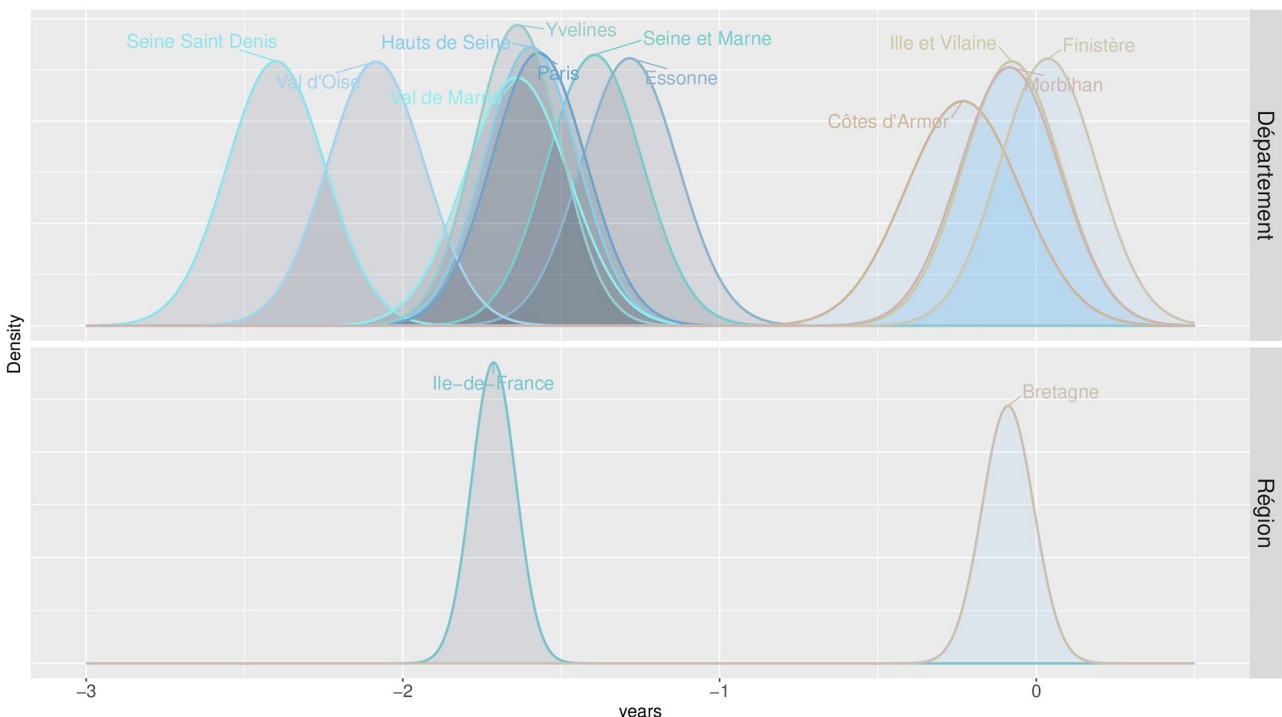

**Fig 5. Illustrative figure on the effects of spatial aggregation in excess mortality estimation.** Densities of losses/gains in life expectancy at 60 for two NUTS 2 populations (*Ile-de-France* and *Bretagne*, lower panel) and their associated NUTS 3 populations (upper panel).

[1.57 – 1.85]. Still, this result hides strong heterogeneity between the least and the hardest-hit *département* in this *région*: *Esonne* with [0.98 – 1.68] and *Seine-Saint-Denis* with [2.10 – 2.70]. Even before 2020, *Seine-Saint-Denis* had one of the highest mortality levels in France. Concealing its further pandemic-related deterioration through spatial aggregation to reduce uncertainty would be inappropriate from a health policy perspective. In contrast, estimates for *Bretagne* (Brittany) do not mask significant spatial heterogeneity, and aggregation within this region considerably reduces the confidence interval around the excess mortality estimates without much loss of information.

An alternative method to reduce uncertainty while maintaining the same administrative division would be to combine men and women. This strategy would practically double the population sizes and reduce the associated 95% confidence intervals by 24%. However, estimates for both sexes will obscure the heterogeneity between men and women, especially considering the substantial sex differences in COVID-19 morbidity and mortality highlighted in the literature [44, 45]. To ensure comprehensive coverage, S3 Appendix presents figures that depict estimates and their corresponding uncertainties, taking into account spatial aggregation at the NUTS-2 level and aggregation across both sexes.

Unlike previous methods for estimating excess mortality, ours effectively addresses all challenges associated with small populations. Its robustness, flexibility, and low computational cost make it highly suitable for mapping the impact of COVID-19 at the international level. We encourage national statistical offices to expedite the publication of regional-level mortality data, which, coupled with our available routines, would enable a more accurate and timely assessment of the burden of any ongoing pandemic.

## Supporting information

**S1 Appendix. Method for computing excess mortality.** Presents derivations for computing, from *CP*-spline estimated coefficients and observed mortality rates, excess mortality measured by different demographic indicators.
(PDF)

**S2 Appendix. Comparison with the Lee-Carter model.** Presents a comparison between *CP*-spline and the Lee-Carter model of excess mortality estimates.
(PDF)

**S3 Appendix. Additional figures and tables.** Presents additional figures, maps and tables of excess mortality estimates.
(PDF)

## Acknowledgments

The authors would like to thank the three anonymous reviewers as well as researchers of "Mortality, Health, Epidemiology" Research Unit at INED (Aubervilliers, France), researchers of REDIM team at BiB (Wiesbaden, Germany) and participants of the European Population Conference 2022 for their invaluable comments.

## Author Contributions

**Conceptualization:** Florian Bonnet, Carlo-Giovanni Camarda.

**Data curation:** Florian Bonnet, Carlo-Giovanni Camarda.

**Formal analysis:** Florian Bonnet, Carlo-Giovanni Camarda.

**Funding acquisition:** Florian Bonnet, Carlo-Giovanni Camarda.

**Investigation:** Florian Bonnet, Carlo-Giovanni Camarda.

**Methodology:** Florian Bonnet, Carlo-Giovanni Camarda.

**Project administration:** Florian Bonnet, Carlo-Giovanni Camarda.

**Resources:** Florian Bonnet, Carlo-Giovanni Camarda.

**Software:** Florian Bonnet, Carlo-Giovanni Camarda.

**Supervision:** Florian Bonnet, Carlo-Giovanni Camarda.

**Validation:** Florian Bonnet, Carlo-Giovanni Camarda.

**Visualization:** Florian Bonnet, Carlo-Giovanni Camarda.

**Writing – original draft:** Florian Bonnet, Carlo-Giovanni Camarda.

**Writing – review & editing:** Florian Bonnet, Carlo-Giovanni Camarda.

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
