## [Decision Letter · Decision Letter 0]

28 Jun 2023

PONE-D-23-16951Estimating Subnational Excess Mortality in Times of Pandemic. An application to French départements in 2020PLOS ONE

Dear Dr. Bonnet,

Thank you for submitting your manuscript to PLOS ONE. After careful consideration, we feel that it has merit but does not fully meet PLOS ONE’s publication criteria as it currently stands. Therefore, we invite you to submit a revised version of the manuscript that addresses the points raised during the review process.

We look forward to receiving your revised manuscript.

Kind regards,

Bruno Ventelou

Academic Editor

PLOS ONE

Journal Requirements:

"The funders had no role in study design, data collection and analysis, decision to publish, or preparation of the manuscript"

4. We note that Figure 3 in your submission contain [map/satellite] images which may be copyrighted. All PLOS content is published under the Creative Commons Attribution License (CC BY 4.0), which means that the manuscript, images, and Supporting Information files will be freely available online, and any third party is permitted to access, download, copy, distribute, and use these materials in any way, even commercially, with proper attribution. For these reasons, we cannot publish previously copyrighted maps or satellite images created using proprietary data, such as Google software (Google Maps, Street View, and Earth). For more information, see our copyright guidelines: http://journals.plos.org/plosone/s/licenses-and-copyright.

a. You may seek permission from the original copyright holder of Figure 3 to publish the content specifically under the CC BY 4.0 license.  

Additional Editor Comments:

I have received 3 reports from 3 referees. They are raising some issues that you have to deal with, before resubmitting. Please, have a scrupulous attention on these 3 reports below.

Reviewers' comments:

Reviewer's Responses to Questions

**Comments to the Author**

1. Is the manuscript technically sound, and do the data support the conclusions?

Reviewer #1: Partly

Reviewer #2: Yes

Reviewer #3: Partly

2. Has the statistical analysis been performed appropriately and rigorously? 

Reviewer #1: I Don't Know

Reviewer #2: Yes

Reviewer #3: Yes

3. Have the authors made all data underlying the findings in their manuscript fully available?

Reviewer #1: Yes

Reviewer #2: Yes

Reviewer #3: Yes

4. Is the manuscript presented in an intelligible fashion and written in standard English?

Reviewer #1: Yes

Reviewer #2: Yes

Reviewer #3: No

5. Review Comments to the Author

Reviewer #1: On “Estimating Subnational Excess Mortality in Times of Pandemic…”

Even though the epidemic is over, the issue of the losses caused by it, in particular its detailed analysis, remains important. The paper does a very good job of explaining the regional variation in excess mortality during the COVID period in France and can make a very good contribution to the field.

The general approach of the paper (based on full mortality projections rather than base year figures or simplified extrapolations of deaths) is sound, the text is well written.

However, my main concern is with the CP-splines used in the paper. Apart from general concerns about splines used to extrapolate beyond the data to which they are fitted, the extrapolations by the CP splines are partly based on ad hoc solutions, such as limiting the rate of change in mortality to 50% of its variation in the past. This could, for example, lead to overly pessimistic projections of mortality decline. It could also reduce estimates of the uncertainty of the prediction (in what I read about the method, I did not see discussion of whether the ad hoc constraints imposed on the model might affect the uncertainty; please check this point). I suggest checking the validity of CP-splines based extrapolations by comparing them, at least in some cases, with a simpler, more transparent method, such as linear extrapolation of log-mortality rates, or the technically similar Lee-Carter method. If the methods appear to give similar results in terms of excess mortality for selected regions, this would be a good support for your methodology, especially for those readers for whom the CP splines are too technical a model. In any case, you might need to explain the basics of CP-splines and their use for extrapolation in more detail, but in simple language; discussing possible limitations of the method and its estimated uncertainty.

Finally, the discussion is too weak for such a topical paper as yours. I suggest that you provide a more detailed substantive discussion of what might explain the regional differences and perhaps put your results in the context of previous publications for France and other countries.

Reviewer #2: This paper presents a method for estimating excess pandemic mortality on a sub-national scale. The authors present an application of their method to Covid-19 mortality data in France, at département level, in order to estimate excess Covid mortality in 2020. The paper is well produced overall, but a few imprecisions persist in the methodology presented and in some of the results, which would benefit from clarification. Here are a few comments:

- Section Methods, page 2, line 67: "However, without loss of generality, computation of excess mortality in successive years can be obtained by extending the forecast horizon presented below."

It seems to me that by extending the forecasting period in this way, the estimates for the following years (2021+) would be somewhat deteriorated, as the French government's reaction to covid has not been constant throughout this period (mainly in 2020: containment and closure, in 2021: health pass and vaccine) and this may have a direct effect on mortality. The authors should therefore present the results at least for the year 2021, discussing both the quality of the forecasting and giving some hints of interpretation in terms of public policy and health context (appearance of new waves of variants).

- Section Methods, page 2, line 67: "For a given subpopulation and sex"

The authors should clarify early in the paper (also in the abstract) that the aim is to obtain a separate excess mortality estimator for men and women in each département. Furthermore, why differentiate between subpopulation and gender? While gender is a discriminating criterion, this needs to be clarified.

Furthermore, all the results presented are limited to the excess mortality in men, why this choice? It seems to me that this is not discussed in the paper; the idea of combining men and women to obtain less uncertainty is well discussed, as is the heterogeneity in mortality between men and women. The authors should give more reasons for this choice.

- Section Methods, page 3, line 73: "Number of deaths d_{ij} at age i in year j are assumed to be realizations from a Poisson distribution"

Authors should add a sentence about the exposure n_{ij}, does it corresponds to the number of individuals at age i in year j ?

- Section Methods, page 3, line 84: "where B a two-dimensional model matrix that combines B-splines over age and years."

Although the paper makes a reference to "Camarda CG. Smooth Constrained Mortality Forecasting. Demographic Research. 2019;41(38):1091–1130. doi:10.4054/DemRes.2019.41.38." authors should give more insight about the way of computing this matrix and its relations with vectors d and n, or maybe add a supplementary mathematical section.

- Section Methods, page 3, line 98:

What is e_{2020} ? The denominator of the observed death rate ?

- Section Methods, page 3, equation 2:

The notation "e" is used here to represent the life expectancy at a given age, is it the same "e" referring to my previous question ? This should be clarified.

- Figure 2 & 3:

Looking at the map of French departments, there doesn't seem to be any significant gain, but figure 2 seems to say the opposite for the "Gers" department. Why does this department appear significant in figure 2 when it seems to be the same indicator shown in figure 3?

Overall, the article has a strong public health interest. Although there are still a few details to be taken into account and clarified, it informs and documents the literature on excess mortality in times of pandemic, with a well-executed application to the French départements.

Reviewer #3: This study estimates the loss of life expectancy in 2020 in 95 French departments. The study has qualities that are unfortunately masked by the fact that the paper is poorly written and structured. The paper's qualities are: 1) to give a picture of the sub-regional distribution of the impact of the pandemic in France 2020, 2) to use a sound method (already published) to project pre-pandemic mortality in 2020, and 3) to express results in terms of life expectancy, an important and intuitive metric.

Unfortunately, the article in its current form puts too much emphasis on the method used, with too little comment on the results themselves, treated as a kind of "example" of applying the methodology rather than being the main result of the article. In addition, the writing in English is very poor and needs to be reviewed throughout the paper.

The article should therefore be reworked to focus more on the results and less on the method (which is the subject of an earlier publication) and proofread by a native English speaker.

Introduction:

More information and references are needed on the following three dimensions: geographical granularity (country vs. regions or departments), the way in which excess mortality is estimated (compared with a past level or with estimated past trends, and in the latter case, which model is used), and the metric used (life expectancy, vs. standardized mortality rates or numbers of deaths). If it's true that your method would also apply to other scales, you choose to represent the results in terms of lost life expectancy. This choice is not insignificant. Indeed, it is not common to see studies estimating life expectancy losses taking into account past trends (e.g. Aburto (2022) simply compared 2020 and 2021 to 2019), and this should be emphasized more. On the other hand, even if your model is sound, well described and tested in a previous study, other models have also been applied in the literature, many of which also estimate uncertainty around predicted mortality, and you should mention these other methods as well.

Please refrain from ditirambic statements about the model you've chosen, such as the one in line 52: “we present a fully analytic procedure with enormous advantages from computational cost and time perspective”. What is a “fully analytic procedure”?

Methods:

The prediction model used is a Poisson model including splines for age and period, with some constraints on the coefficients of the spline basis vectors (CO-splines) to avoid aberrant prediction behavior. Given that the method comes from a previous study (Camarda, 2019), it is definitely not useful to go into so many details in the method Section. For example, on line 81, “let arrange the complete matrix in column vectors ….”. I would also avoid formulas as the one in equations (2-4), which are at the time too complex and necessarily incomplete, as they rely for example on a (mysterious) matrix L : ”constructed in order to select 2020 forecast mortality from (1)”. Since all these technical details have previously been described in previous papers, I strongly recommend to simply refer to the literature for more detailed explanations.

Again, avoid some ditirambic statements about the model, as in line 77: “This method allows us to simultaneously estimate and forecast mortality within a regression setting with enormous advantages in the computation of uncertainty measures”. I would remind that every prediction method will give simultaneously estimation and predictions of future mortality, and any Poisson model will allow to estimate uncertainty around predictions, which is certainly not a special feature of CP-splines.

I think it's a good idea to use cross-validation to select the time window for estimation. However, why set the minimum window at 2010-2018/19? I'd also like to see what range is obtained for this window across the departments.

Data and applications

First, I would call this section “Results”.

Figure 1 is quite interesting as an example. However, I would limit myself to the two lower panels, which give your complete results for this department. Here, the comparison with a factual pre-pandemic level is actually interesting, but I'd prefer to see it on the lower panels, where the 95%CI around life expectancy loss actually includes the loss relative to the 2015-19 period. This should be highlighted. I'd also prefer a comparison with 2019, rather than 2015-19, since the former choice is the most widely used in the literature (and the most intuitive, given the trend of increasing life expectancy). Then, such a comparison with previous levels is not, in my view, a "naive" comparison, but rather a "factual" one and, in that sense, an (equally) respectable choice. For example, one can state that life expectancy in France in 2020 has fallen by around half a year compared to 2019 (this is not naive, it's just factual), and add that, compared to the trend in life expectancy before the pandemic, the loss has been around 9 months.

Figures 2 and 3 are, for me, the ones that give interest to the paper. Separating the two sources of variability (the model and the data variability) in Figure 2 is a nice way of representing the results. I'd like to see more description of these figures in the text. I'd also like to see the same figures for women and a comparison between the two sexes and between regions in the results section. Can one see (approximately) the same order of the departments for the two sexes (as it seems reasonable) ? Again, in Figure 2 please replace the comparison to 2015-19 with one to 2019, much more effective.

Figure 4 doesn’t add much to the results. The messages of these figure can easily be summarized in the text.

Figure 5 is interesting for me, since it shows how a larger spatial granularity can hidden more fine variability. However, it should be placed and commented in the Results section.

Some results are missing:

1) I understand that you're estimating several models, one for each department and gender. However, at least for France and perhaps for some "extreme" departments such as Ile de France and Bretagne, I'd really like to see in one (or more) additional figure(s) the fit of the model at the scale of life expectancy (at age 60), with the observed series, the estimated series and their projections for 2020. Especially when using quite complex and adaptive methods, the fit of a model remains one important piece of information.

2) In addition, it would be extremely important to report in the Annexes tables, not only the estimated life expectancy losses in each department, but instead: 1) the life expectancy in 2019, 2) the life expectancy in 2020, 3) the predicted life expectancy in 2020, 4) 2020-2019 (instead of what you call “naïve” comparison w-r-t 2015-2019) and 5) 2020 - predicted 2020 (with your confidence intervals). Was the life expectancy level quite uniform across departments in 2019? Is the loss depending on the initial level? Are results similar for men and women?

Discussion

A more in-depth discussion of your results for the French departments should be added, with a comparison between regions and sexes, and some attempts to explain the observed variability.

Since all your results are based on life expectancy (losses) at age 60, a few words on life expectancy at birth should be added. Is there any change in your results when considering life expectancy at birth?

6. PLOS authors have the option to publish the peer review history of their article (what does this mean?). If published, this will include your full peer review and any attached files.

Reviewer #1: No

Reviewer #2: **Yes: **Pierre MICHEL

Reviewer #3: No

---

## [Decision Letter · Decision Letter 1]

18 Oct 2023

Estimating Subnational Excess Mortality in Times of Pandemic. An application to French départements in 2020

PONE-D-23-16951R1

Dear Dr. Bonnet,

We’re pleased to inform you that your manuscript has been judged scientifically suitable for publication and will be formally accepted for publication once it meets all outstanding technical requirements.

Kind regards,

Bruno Ventelou

Academic Editor

PLOS ONE

Additional Editor Comments (optional):

Congratulations, your paper is accepted

Just, I tranfer a remark from one of the two referees:

"a minor comment, more for your future work, I 'd like to point out that you seem to compare your method to the LC variant with no jump in rates in the base year, which seems to explain some of the differences you present, while your extrapolation does care about such jumps. Depending on how you feel about the plausibility of such jumps, you may lean towards one method or the other - but because of the different treatment of the jump issue, not because of the inherent methodological differences between CP and LC. Apart from that, the age patterns are indeed more plausible in your method"

Reviewers' comments:

Reviewer's Responses to Questions

**Comments to the Author**

1. If the authors have adequately addressed your comments raised in a previous round of review and you feel that this manuscript is now acceptable for publication, you may indicate that here to bypass the “Comments to the Author” section, enter your conflict of interest statement in the “Confidential to Editor” section, and submit your "Accept" recommendation.

Reviewer #1: All comments have been addressed

Reviewer #2: All comments have been addressed

2. Is the manuscript technically sound, and do the data support the conclusions?

Reviewer #1: Yes

Reviewer #2: (No Response)

3. Has the statistical analysis been performed appropriately and rigorously? 

Reviewer #1: Yes

Reviewer #2: Yes

4. Have the authors made all data underlying the findings in their manuscript fully available?

Reviewer #1: Yes

Reviewer #2: (No Response)

5. Is the manuscript presented in an intelligible fashion and written in standard English?

Reviewer #1: Yes

Reviewer #2: Yes

6. Review Comments to the Author

Reviewer #1: I find the revised paper, especially with the useful and instructive CP-LC comparison, solid and ready to go out.

As a minor comment, more for your future work, I 'd like to point out that you seem to compare your method to the LC variant with no jump in rates in the base year, which seems to explain some of the differences you present, while your extrapolation does care about such jumps. Depending on how you feel about the plausibility of such jumps, you may lean towards one method or the other - but because of the different treatment of the jump issue, not because of the inherent methodological differences between CP and LC. Apart from that, the age patterns are indeed more plausible in your method.

Reviewer #2: I would like to thank the authors for responding to all my comments in order to clarify their paper.

The authors have answered my comments correctly, providing the methodological details necessary for the reader to understand the paper. In addition, the authors' responses and the new additions to the paper help to clarify the methodological points and also the objectives of the paper.

Additional results have been added in Supporting Information to get excess mortality estimators for women.

Clarifications have been made to the methods section, in particular to the mathematical notations which were confusing. In particular, the notion of exposure is now described in the method, making the rest of the paper easier to read, as is the construction of 2-dimensional B-splodes by adding some references in the bibliography.

Errors of notation have been corrected, as have errors in Figures 2 and 3.

7. PLOS authors have the option to publish the peer review history of their article (what does this mean?). If published, this will include your full peer review and any attached files.

Reviewer #1: No

Reviewer #2: No

---

## [Editor Report · Acceptance letter]

3 Nov 2023

PONE-D-23-16951R1 

Estimating Subnational Excess Mortality in Times of Pandemic. An application to French *départements* in 2020 

Dear Dr. Bonnet:

I'm pleased to inform you that your manuscript has been deemed suitable for publication in PLOS ONE. Congratulations! Your manuscript is now with our production department. 

Kind regards, 

on behalf of

Dr. Bruno Ventelou 

Academic Editor

PLOS ONE